# Health systems interventions for hypertension management and associated outcomes in Sub-Saharan Africa: A systematic review

Samuel Byiringiro [1] *, Oluwabunmi Ogungbe [1], Yvonne Commodore-Mensah [1,2], Khadijat Adeleye [3], Fred Stephen Sarfo [4,5], Cheryl R. Himmelfarb [1,2]

1 Johns Hopkins University School of Nursing, Baltimore, Maryland, United States of America, 2 Johns Hopkins University, Bloomberg School of Public Health, Baltimore, Maryland, United States of America, 3 University of Massachusetts, Amherst, MA, United States of America, 4 Department of Medicine, Kwame Nkrumah University of Science & Technology, Kumasi, Ashanti Region, Ghana, 5 Komfo Anokye Teaching Hospital, Kumasi, Ashanti Region, Ghana

* sbyirin1@jhu.edu

**Data Availability Statement:** All data in manuscript and supporting information files.

## Abstract

Hypertension is a significant global health problem, particularly in Sub-Saharan Africa (SSA). Despite the effectiveness of medications and lifestyle interventions in reducing blood pressure, shortfalls across health systems continue to impede progress in achieving optimal hypertension control rates. The current review explores the health system interventions on hypertension management and associated outcomes in SSA. The World Health Organization health systems framework guided the literature search and discussion of findings. We searched PubMed, CINAHL, and Embase databases for studies published between January 2010 and October 2022 and followed the Preferred Reporting Items for Systematic Reviews and Meta-Analyses guidelines. We assessed studies for the risk of bias using the tools from the Joanna Briggs Institute. Twelve studies clustered in 8 SSA countries met the inclusion criteria. Two thirds (8/12) of the included studies had low risk of bias. Most interventions focused on health workforce factors such as providers' knowledge and task shifting of hypertension care to unconventional health professionals (n = 10). Other health systems interventions addressed the supply and availability of medical products and technology (n = 5) and health information systems (n = 5); while fewer interventions sought to improve financing (n = 3), service delivery (n = 1), and leadership and governance (n = 1) aspects of the health systems. Health systems interventions showed varied effects on blood pressure outcomes but interventions targeting multiple aspects of health systems were likely associated with improved blood pressure outcomes. The general limitations of the overall body of literature was that studies were likely small, with short duration, and underpowered. **In conclusion**, the literature on health systems internventions addressing hypertension care are limited in quantity and quality. Future studies that are adequately powered should test the effect of multi-faceted health system interventions on hypertension outcomes with a special focus on financing, leadership and governance, and service delivery interventions since these aspects were least explored.

**Funding:** The authors received no specific funding for this work.

**Competing interests:** The authors have declared that no competing interests exist.

**Abbreviations:** CINAHL, Cumulated Index to Nursing and Allied Health Literature; JBI, Joanna Briggs Institute; HIV, Human Immunodeficiency Virus; LMIC, Low- and Middle-Income Countries; PRISMA, Preferred Reporting Items for Systematic Reviews and Meta-Analyses; SARA, Service Availability and Readiness Assessment; SDG, Sustainable Development Goals; COVID-19, Coronavirus Disease 2019; SQUIRES, Standards for Quality Improvement Reporting Excellence; SSA, Sub-Saharan Africa; WHO, World Health Organization.

## Introduction

Hypertension remains a global health challenge, especially for low- and middle-income countries (LMICs) in Sub-Saharan Africa (SSA) [1]. Hypertension is a significant risk factor for stroke and ischemic heart disease, the world's deadliest maladies [2]. It is estimated that 30% of the SSA population have hypertension [3–5], 27% are aware of their hypertensive status, and only 7% have their hypertension under control [3]. The prevalence of hypertension in SSA will likely continue to rise due to the rapid shift to sedentary lifestyles, growing urbanization, and a higher number of people living longer [6, 7]. The level of preparedness of health systems for managing hypertension and related sequelae is poor and requires urgent attention [8, 9].

Several strategies have been implemented to manage hypertension in SSA. Population-based interventions such as educational campaigns to improve dietary practices by reducing salt consumption have proven cost-effective in lowering blood pressure (BP) among hypertensive patients [10–13]. Alternative interventions implemented to reduce dietary salt include increasing the availability and affordability of healthy foods and reformulation of processed foods with the help of government policies [10]. Other strategies, including technology and non-technology-based reminders, targeted patients with hypertension to improve their adherence to the prescribed pharmacologic and non-pharmacologic treatments [13, 14]. Despite the numerous effective interventions targeting patients with hypertension, the interventions aimed at improving the health systems' capacity to provide high-quality hypertension care in SSA are scarce [8, 9].

Multiple guidelines exist for screening, diagnosing, and treating hypertension [2, 15, 16]. The Pan-African Society of Cardiology task force convened in 2014 laid the roadmap to achieving 25% BP control in Africa by 2025 [17]. In 2021, the World Heart Federation laid an updated roadmap that identified the major roadblocks in hypertension diagnosis and management and recommended solutions for reducing uncontrolled hypertension by 30% by 2030 [18]. The lack of local hypertension treatment guidelines, protocols, and policies on the procurement and distribution of anti-hypertensive medications are some of the multiple identified government and health system roadblocks to achieving hypertension control goals [17, 18]. These roadblocks likely result in the lack of community sensitization on hypertension and service inaccessibility through high-cost and centralized hypertension care [19, 20]. According to the Chronic Care Model, the health systems' role in hypertension management should be to support patients for self-management at home, set up systems to ensure patients follow up after clinical encounters, and decentralize hypertension care closest to patients [20, 21]. Health systems should provide access to specialists if needed, ensure providers are trained and have access to the latest guidelines and decision support, and implement strong health information management to improve the quality of care [20, 21].

The World Health Organization (WHO) health systems framework highlights six critical elements to target while working to improve the quality of health services [22]. These are service delivery; health workforce; medical products and technology; health information systems; financing; and leadership and governance. Using the WHO health systems framework, the current review synthesizes the limited literature reporting on the effect of health systems interventions on hypertension and associated outcomes in SSA.

## Methods

### Search strategy

Collaborating with a scientific library informationist, we defined the search terms for hypertension service availability and delivery guided by the WHO health systems' framework [22].

We Identified Medical Subject Headings (MeSH) terms ("Health Facilities," "Health Services Accessibility," "Health Care Delivery"); and hypertension ("hypertension" or "elevated blood pressure" or "raised blood pressure" or anti-hypertensive) in SSA. We searched PubMed, Cumulated Index to Nursing and Allied Health Literature (CINAHL), and Embase (S1 Table).

## Study selection and eligibility

The Preferred Reporting Items for Systematic Reviews and Meta-Analyses (PRISMA) [23] guided this systematic review (S1 Checklist). We imported articles from the literature search into Covidence, and three investigators (SB, KA, and OO) conducted the title and abstract screening, full-text review, and final extraction. Two investigators appraised each article independently at each step, and the third investigator resolved any conflicts. Occasionally, the investigators discussed with the supervising faculty (YCM and CRH) to resolve disputes.

In the article search and screening, we included studies (1) published in English or French between January 2010 –October 2022; (2) conducted in health facility settings (hospitals and clinics) at different levels of the health system in SSA; (3) with a randomized controlled trial or quasi-experimental design; (5) exploring the impact of health systems-level interventions on any of the following hypertension outcomes: awareness, treatment initiation and adherence, systolic and diastolic BP levels, and achievement of BP control.

We excluded (1) observational studies and (2) studies that explored hypertension care interventions conducted solely in the community setting without describing collaboration with health facilities, or (3) examining solely patient-level behavioral change interventions for hypertension management. Additionally, we excluded gray literature (books, cases, etc.).

The investigators tracked the rationale for excluding any study during the entire process following the article search.

**Applying the WHO health systems framework.**   To synthesize our findings, we examined each study in view of the six WHO key building blocks of the health systems: service delivery; health workforce; medical products and technologies; information systems; financing; and leadership and governance [22].

*Service delivery.* WHO defines service delivery as the activities which result in the direct provision of safe, effective, and high-quality health services to those who need them. We considered the studies to address the service delivery aspect of the health systems if the target of the intervention was to make hypertension services (screening, treatment, and follow-up) more accessible through decentralization of care, reduction of waiting time, revision of time of services, or application of services integration delivery.

*Health workforce.* In the health workforce aspect of the health systems, we included studies with interventions that targeted the healthcare providers who manage patients with hypertension. The interventions included strategies to improve provider density, increase provider knowledge and adherence to hypertension guidelines, address provider decision support systems, promote teamwork, and institute task sharing and/or task shifting strategies among providers who do not normally perform certain hypertension management tasks.

*The medical products and technologies.* The interventions of interest regarding the medical products and technologies were the procurement systems to ensure the availability of anti-hypertensive medications, calibrated and validated BP measurement devices (as well as their maintenance), technologies for easier patient management and follow-up, hypertension treatment guidelines, and consumables for hypertension-related laboratory exams.

*Health information systems.* We considered interventions to address health information systems if they studied an element of the process of patient data collection, analysis, sharing, and use to improve outcomes of patients with hypertension. We included interventions that

examined the use of patient registries and patient-to-provider and provider-to-provider information sharing in patient management.

*Financing.* The financing aspect of the health systems, as defined in the WHO building blocks, has the most significant relevance on the national macro-level health systems yet plays an impact on the micro-level health systems' service delivery and patients' outcomes. In the current context, we considered the interventions which evaluated the injection of money into the health systems through health insurance premiums, reduction of the patients' out-of-pocket spending, or funding for hypertension management activities at the health facilities.

*Leadership and governance.* Leadership and governance have direct and indirect associations with the other five building blocks of the health systems. In this review, we considered an intervention to target leadership and governance if it promoted leadership awareness of the rising burden of hypertension, used strategic planning or implementation of national strategy around hypertension management activities, explored the accountability measures at the health facility, applied regular performance appraisal and planning for improvement, integrated supportive mentorship to low health system level health facilities, instituted a patient feedback collection and response system, or studied the leadership allocation and reallocation of funds for hypertension management. We also considered interventions around leadership joining collaborations (with other health facilities or local/international/national/private organizations) to manage hypertension.

**Risk of bias assessment.** We used the Joanna Briggs Institute (JBI) tools to assess the quality aspects of the methods and conduct of the studies included in the review [24]. We evaluated the one intervention that utilized a quality improvement strategy using the Standards for Quality Improvement Reporting Excellence 2.0 (SQUIRES) guideline [25]. SQUIRES uses a checklist of standard elements for reports of system-level projects to improve healthcare quality. Two investigators independently assessed the risk of bias for each study, and the third resolved conflicts. The JBI tool uses "Yes," "No," and "Unclear or Not applicable" to assess the different features of study design and the overall appraisal as include study, exclude study, or seek more information. We replaced "Yes," "No," and "Unclear" with "Low risk of bias," high risk of bias," and "Some concerns," respectively, for each statement on the JBI tool. We assigned the overall quality appraisal of each study as "good," "fair," and "poor" and studies with poor quality were excluded. We used the robvis [26] online tool to create risk-of-bias plots where the overall appraisal was also converted to "low risk of bias" for the study of good quality, "some concerns" for studies of fair quality and "high risk of bias" for studies of poor quality.

**Analysis and synthesis of findings.** After the data extraction, we identified the health systems factors addressed by the interventions for hypertension care. We classified the outcomes of each study into the appropriate group: hypertension awareness, systolic and diastolic BP, treatment initiation and adherence, and BP control. We tracked the reported effect(s) of interventions on hypertension outcomes and compared findings across the different countries of SSA.

## Results

We identified 7,494 studies and 3,121 were duplicates (Fig 1). We conducted title and abstract screening for 4,373 studies yielding 205 for full-text review. The latter led to 12 studies for inclusion in this systematic review (Table 1). Of the 12 included studies, six were randomized controlled trials conducted in South Africa (n = 2) [27, 28], Cameroon (n = 1) [29], Ghana (n = 1) [30], Kenya and Uganda (n = 1) [31], and Zambia (n = 1) [32]. Six quasi-experimental studies were conducted in Nigeria (n = 2) [33, 34], Botswana (n = 1) [35], Cameroon (n = 1) [36], Kenya (n = 1) [37], and Sierra Leone (n = 1) [38]. The geographic representation of included studies by country is demonstrated in Fig 2.

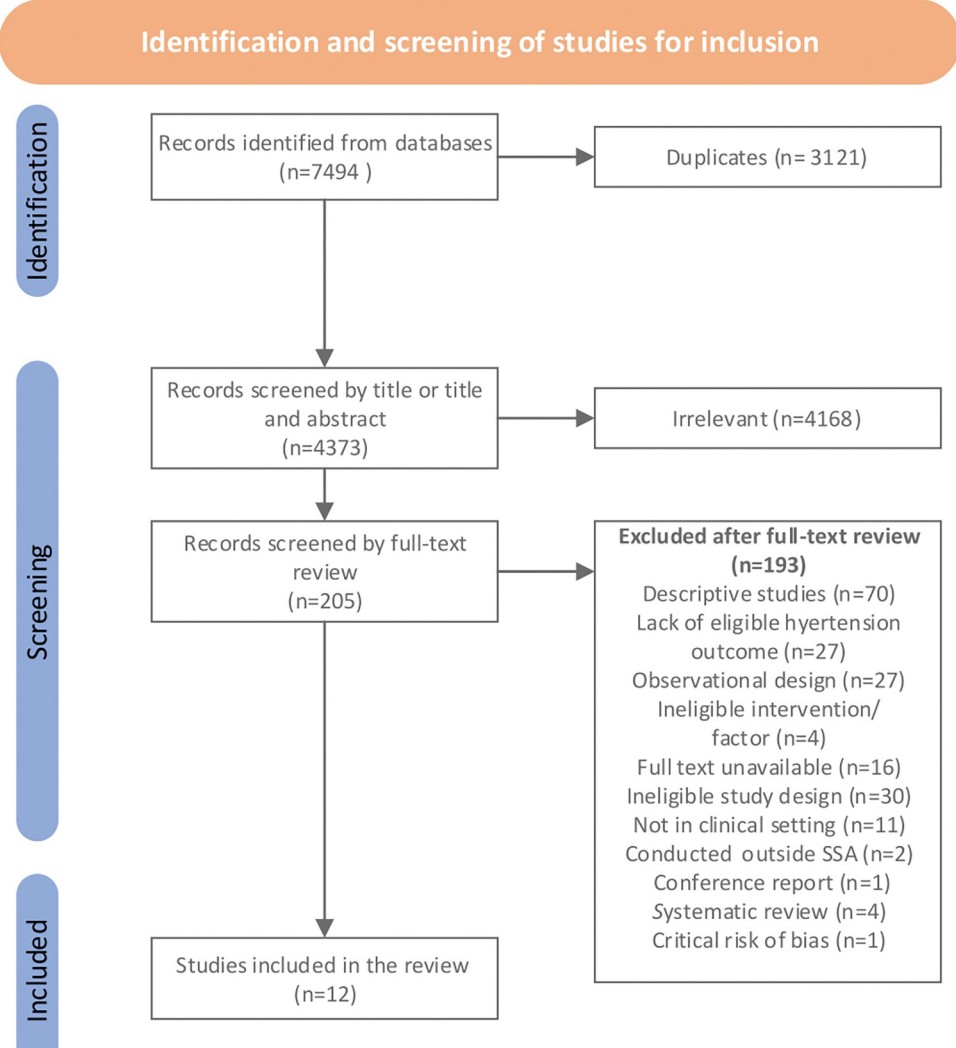

**Fig 1. PRISMA diagram with the flowchart of study screening and inclusion.**

The quality assessment demonstrated a low risk of bias in 67% (8/12) of the included studies (S1 Fig & S2 Table). Among randomized controlled trials, the most common issues were the lack of: concealed intervention allocation, similarity of groups at baseline, and blinding participants and those delivering the intervention about the group (intervention or control groups) assignments. Though it is not feasible to conceal intervention allocation nor to blind participants or interventionists in such health system interventions, the adequate description of the reason for omitting standard procedures of a study design is desirable.

## Health system factors and interventions explored

The health system factor that most interventions targeted is the health workforce, followed by interventions to address equipment and medicines, information systems, financing, service delivery, and leadership and governance (Table 2). The detailed findings from included papers can be found in Table 3.

**1. Service delivery.** Only one study conducted in Kenya and Uganda addressed the service delivery aspect of the health systems on hypertension care [31]. The study explored the cross-

**Table 1. Characteristics of studies examining the impact of health system interventions on hypertension outcomes (N = 12).**

| Study | Country | Study Design | Population and Sample size | Description of intervention | Health system factors/interventions explored | | | | | |
|---|---|---|---|---|---|---|---|---|---|---|
| | | | | | Service Delivery | Health workforce | Medical products and tech. | Information systems | Financing | Leadership & Governance |
| Goudge, 2018 [28] | South Africa | RCT | 3978 patients at baseline receiving care at eight community health clinics | Applied task-shifting strategy by which lay health workers supported the management of hypertension. | | ★TS | | | | |
| Hendriks, 2014 [33] | Nigeria | Quasi-experimental | 1500 patients with hypertension | The participants in the intervention group received enrollment in the health insurance program, while those in the control arm did not. | | | | | ★HIP | |
| Hickey, 2021 [31] | Kenya and Uganda | RCT | 86,078 adults with uncontrolled hypertension | Intervention sites integrated HIV and hypertension care, followed by in-person or phone-based follow-up if the patient misses an appointment. | ★ | ★TS | | | | |
| Kande, 2014 [35] | Botswana | Pre & post-quasi-experimental QI Project | 233 people from the general population attending the health clinic | Established an audit team, set standards of care, did monitoring through regular data collection and analysis, and procured missing equipment. | | | ★ | ★ | | |
| Kingue, 2013 [29] | Cameroon | RCT | 268 patients with hypertension who receive care at 30 health clinics | Intervention sites prescribed only generic anti-hypertensive medications. Phone communication between providers was rendered available to intervention sites for clinicians to liaise with experts about complicated cases. | | ★ | | ★ | | |
| Labhardt, 2010 [36] | Cameroon | Quasi-experimental | 79 primary health facilities | Did training of healthcare providers and task shifting of hypertension care to non-physician clinicians. Participating facilities received equipment for blood pressure and blood sugar screening and an initial stock of anti-hypertensive medications. There was, additionally, monitoring and evaluation of performance. | | ★TS | ★ | | | |

**Table 1.** (Continued)

| Study | Country | Study Design | Population and Sample size | Description of intervention | Health system factors/interventions explored | | | | | |
|---|---|---|---|---|---|---|---|---|---|---|
| | | | | | Service Delivery | Health workforce | Medical products and tech. | Information systems | Financing | Leadership & Governance |
| Nelissen, 2018 [34] | Nigeria | Quasi-experimental | 336 adult patients with hypertension receiving medications at community pharmacies | Applied task shifting of hypertension care to pharmacists and a mobile health application to manage patient information and facilitate sending reminders to patients. Doctors and pharmacists collaborated and received a financial incentive in function to the number of patients consulted | | ★TS | | ★ | ★ | |
| Ogedegbe, 2018 [30] | Ghana | RCT | 640 patients with hypertensive and being followed at community health clinics | Patients in both arms received health insurance coverage, and the intervention sites applied a nurse-led task-shifting strategy | | ★TS | | | ★HIP | |
| Ogola, 2019 [37] | Kenya | Quasi-experimental | 132 health facilities | Provided training of trainers to consultant physicians and nurses who, in return, trained their local healthcare professionals in their respective regions on managing hypertension. Provided equipment and patient educational materials and supported the supply of anti-hypertensive medications. | | ★ | ★ | | | |
| Steyn, 2013 [27] | South Africa | RCT | 920 patients with hypertension and diabetes | Structured recording of patient information per the national hypertension guidelines requirements. Inclusion of a copy of guidelines in the register for an easy guide to clinicians. Trained providers on the use of that recording system. | | ★ | | ★ | | |

*(Continued)*

**Table 1.** (Continued)

| Study | Country | Study Design | Population and Sample size | Description of intervention | Health system factors/interventions explored | | | | | |
|---|---|---|---|---|---|---|---|---|---|---|
| | | | | | Service Delivery | Health workforce | Medical products and tech. | Information systems | Financing | Leadership & Governance |
| Yan, 2017 [32] | Zambia | RCT | 35052 patients with hypertension | Standardization of protocols for hypertension care, use of electronic medical records, and ongoing mentoring. Trained community volunteers on and tasked them with vital signs measurement. | | ★TS | ★ | ★ | | |
| Zou, 2020 [38] | Sierra Leone | Quasi-experimental | 50 persons with hypertension and diabetes diagnosis | A multi-stakeholder consortium adapted hypertension guidelines and trained community health officers and doctors who, in turn, trained lower-level healthcare providers in the screening, diagnosis, and management of non-communicable diseases. | | ★TS | ★ | | | ★ |

Abbreviations: **RCT**—Randomized Controlled Trial; **QI**–Quality Improvement

★: **Star represents health system factor addressed; TS**—Task shifting strategy; **HIP**—Health Insurance Premium

integration of HIV and hypertension care services and was associated with 26% (aRR 1.26; p:0.002, 95% CI 1.11, 1.42) higher likelihood of linkage to hypertension care and 29% (aRR 1.29; p < 0.001, 95% CI: 1.03–1.63) higher likelihood of achievement of BP control [31].

**2. Health workforce.** Ten of the twelve included studies evaluated the interventions aimed toward the health workforce [27–32, 34, 36–38]. All ten interventions trained healthcare providers on hypertension treatment guidelines, and seven of them integrated task-shifting strategy [28, 30–32, 34, 36, 38]. Some examples of task-shifting strategy studied are training and giving roles to lay community health workers in South Africa [28] and Zambia [32] to measure BP, having nurses in Cameroon [36] and Ghana [30] and pharmacists in Nigeria [34] to treat hypertension (including assignment of diagnosis and prescribing medications).

Generally, studies that included interventions for the health workforce reported improvement in hypertension outcomes. Improvement in the healthcare providers' skills and knowledge of the correct BP measurement techniques and hypertension diagnosis and treatment, respectively, likely contributed to improved hypertension outcomes in Kenya [37]. In its first year, the program screened 532 527 individuals, diagnosed almost 10% with hypertension, and initiated 72% of those diagnosed on treatment [37].

The impact of task-shifting varied across studies but bent towards positive BP outcomes. In Ghana [30] and Cameroon [36], task-shifting contributed to the significant decline in systolic and diastolic BP but not in the achievement of controlled BP [30, 36]. On the other hand, in the two quasi-experimental studies conducted in Nigeria and Zambia, respectively, the task-shifting contributed to the achievement of controlled BP in Nigeria (AOR 2.27, p: 0.049) from

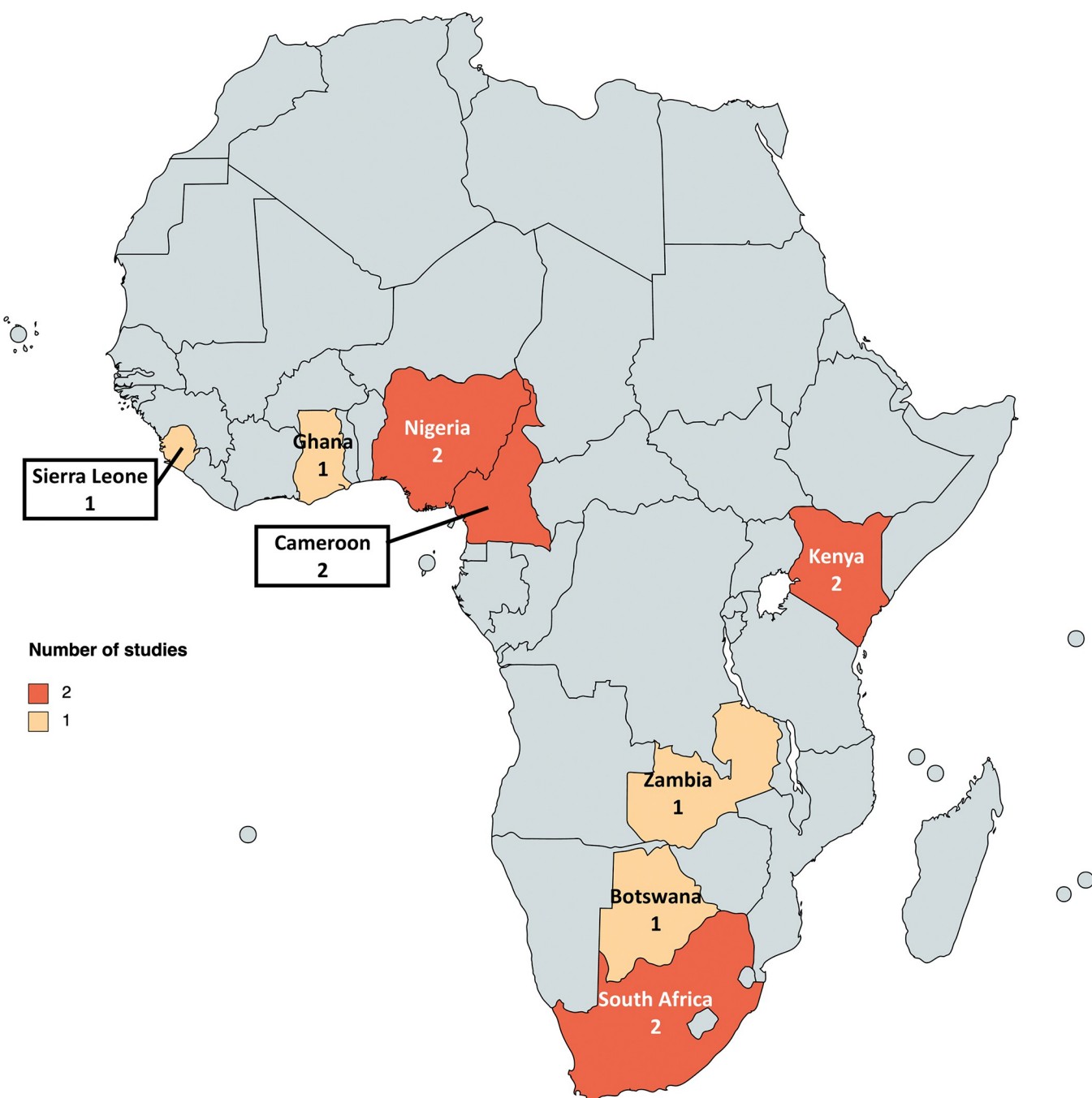

**Fig 2. Distribution of studies on health system interventions on hypertension and associated outcomes in Sub-Saharan Africa.** Base map credit: Andreas 06, Public domain, via Wikimedia Commons (https://commons.wikimedia.org/wiki/File:Blank_Map-Africa.svg).

baseline to end line and in Zambia from 13% at baseline to 25% (no p-value reported) among patients who attended at least two follow up visits [32, 34]. In a cluster randomized controlled trial conducted in South Africa, the task-shifting to lay community health workers in BP measurement and documentation contributed to higher adherence to appointments (75% vs. 56% patients attending their assigned appointments) but no statistically significant improvements in achieving BP control comparing the intervention to the control groups [28].

**Table 2.  The health systems factors addressed by interventions on hypertension and corresponding countries where the interventions were conducted.**

| WHO Health Systems Building Blocks | Number of studies | Country (number of studies) |
|---|---|---|
| Service Delivery | 1 | Kenya and Uganda (1) |
| Health workforce | 10 | South Africa (2), Kenya and Uganda (1), Cameroon (2), Nigeria (1), Ghana (1), Kenya (1), Zambia (1), Sierra Leone (1) |
| Medical products & technologies | 5 | Botswana (1), Cameroon (1), Kenya (1), Zambia (1), Sierra Leone (1) |
| Information systems | 5 | Zambia (1), Cameroon (1), South Africa (1), Botswana (1), Nigeria (1) |
| Financing | 3 | Ghana (1), Nigeria (2) |
| Leadership & Governance | 1 | Sierra Leonne (1) |

Note: Some studies reported more than one system factor

**3. Medical products and technologies.**   Almost half (5/12) of the included studies explored the availability of equipment and consumables involved in hypertension management and the associated care outcomes [32, 35–38]. The procurement of drugs and consumables and improvement in the supply chain implemented in the experimental studies and quality improvement project was in combination with other interventions such as providers' training in Cameroon, Kenya, and Zambia [32, 36, 37], and better data management in Botswana [35]. In these studies, the impact of the equipment and supply chain was not necessarily isolated but contributed to the overall improvements in BP outcomes.

**4. Health information systems.**   Five of the twelve studies evaluated some aspects of health information system interventions on hypertension outcomes [27, 29, 32, 34, 35]. The most common forms of health information system interventions involved patient data recording and analysis for quality improvement [27, 32, 35] and utilization of technology to facilitate patient-provider or provider-provider information exchange for better hypertension outcomes among patients [29, 34]. All interventions that integrated a health information system component demonstrated improved hypertension outcomes except for one randomized controlled trial in South Africa, where hypertension outcomes worsened [27]. In this study, providers reported feeling overwhelmed by the new patient data recording system.

**5. Financing.**   Three studies conducted in Ghana and Nigeria explored the interventions around financing to improve hypertension outcomes[30, 33, 34]. In two studies, the financing was directed towards funding the patients' health insurance premiums in Ghana and Nigeria, respectively [30, 33], while the third study funded performance-based financial incentives to the clinicians who were consulting patients with hypertension in Nigeria [34]. The one quasi-experimental study that provided health insurance coverage to the intervention group reported no change in hypertension awareness, adherence to treatment, or controlled BP [33]. A randomized controlled trial in Nigeria that provided health insurance coverage to both the intervention and control groups reported a decline in systolic and diastolic BP in both study arms, with a significant and sharper decline in the intervention arm even though the achievement of BP control was not significantly different in the intervention versus control arms.

The incentivization of clinicians who manage patients with hypertension contributed to a 66% achievement of BP control among patients even though there was no data for the control group [34].

**6. Leadership and governance.**   While leadership and governance contribute directly or indirectly to all the above health system factors, studies targeting leadership and governance were scarce. A single quasi-experimental study conducted in Sierra Leone explored one aspect

**Table 3. Health systems interventions on hypertension care and associated outcomes (N = 12).**

| Study | Country | Study Design | Health system intervention | Outcomes and key findings | | | |
|---|---|---|---|---|---|---|---|
| | | | | Hypertension awareness | Systolic/Diastolic BP | Hypertension treatment (initiation and adherence to-) | BP Control |
| Goudge, 2018 [28] | South Africa | RCT | Health workforce (TS) | No significant difference between groups. | NR | NR | No significant difference between groups. |
| Hendriks, 2014 [33] | Nigeria | Quasi-experimental | Financing (HIP) | No significant difference between groups. | NR | No significant difference between groups (aOR: 1.37; p = 0.69). | No significant difference between groups AOR 3.16 (95% CI 0.78 to 12.79 p = 0.11). |
| Hickey, 2021 [31] | Kenya and Uganda | RCT | Health workforce (TS) and Service Delivery | Likelihood of patients linkage to hypertension care following BL screening (aRR 1.26; p:0.002, 95% CI 1.11, 1.42). | NR | NR | 56% of intervention group participants and 43% of control group participants achieved hypertension control at year 3 (aRR 1.29; p < 0.001, 95% CI: 1.03–1.63). |
| Kande, 2014 [35] | Botswana | Pre & post quasi-experimental QI Project | Medical products and technology, and Information systems | NR | NR | NR | The goal of achieving 70% BP control from 49% at BL was met, but achieving BP control below 130/80mmHg of 70% from 16% at BL was not met (27% at end-line). |
| Kingue, 2013 [29] | Cameroon | RCT | Health workforce and Information systems | NR | NR | NR | Patients with the third stage of hypertension showed significant achievement of BP control in the intervention compared to control health facilities (39% vs. 50%; p = 0.004). There was no significant difference in BP control for patients with hypertension stages I and II. |
| Labhardt, 2010 [36] | Cameroon | Quasi-experimental | Health workforce (TS) and medical products and technology | NR | Change from BL to 3 months median follow up:Systolic BP: -26.5 mmHg (95% CI: -12.5 to -40.5) Diastolic BP: -17.2 mmHg (95% CI: -7.1 to -27.3). | NR | NR |
| Nelissen, 2018 [34] | Nigeria | Quasi-experimental | Health workforce (TS), Information systems, and financing | NR | NR | Out of 148 patients, 40 showed high adherence, and 39 showed moderate adherence. | 153 of 232 (66%) achieved improved BP. Of patients with high adherence—75.5% achieved improved BP (aOR 2.27 p: 0.049). No significant improvement in BP among patients with moderate adherence to treatment. |

(*Continued*)

**Table 3.** (Continued)

| Study | Country | Study Design | Health system intervention | Outcomes and key findings | | | |
|---|---|---|---|---|---|---|---|
| | | | | Hypertension awareness | Systolic/Diastolic BP | Hypertension treatment (initiation and adherence to-) | BP Control |
| Ogedegbe, 2018 [30] | Ghana | RCT | Health workforce (TS) and Financing (HIP) | | Change in Systolic BP 12 months from BL: 20.4 mm Hg (95% CI: −25.2 to −15.6) in intervention vs 16.8 mm Hg (95% CI −19.2 to −15.6) in the control group. The groups' difference was −3.6 mm Hg (p = 0.021; 95% CI −6.1 to −0.5). | NR | No significant difference between groups. (diff of 5.2%; 95% CI −1.8% to 12.4%). |
| Ogola, 2019 [37] | Kenya | Quasi-experimental | Health workforce and medical products and technology | 532,527 individuals screened for hypertension, and 50,957 diagnosed a new within one year. | NR | 36806 patients of 50957 eligible (72%) initiated on treatment. | NR |
| Steyn, 2013 [27] | South Africa | RCT | Health workforce and Information systems | NR | BP increased from approx. 150/88 to 161/88 mmHg in intervention as in control groups (152/86 to 158/87 mmHg). | NR | In the intervention group, uncontrolled BP increased from 69% to 76%, as in the control group, from 73% to 74%. |
| Yan, 2017 [32] | Zambia | RCT | Health workforce (TS), Medical products and technology, and Information systems | BP screening among health facility users ranged from 65 to 90% at 6 to 48 months after initiation of intervention. The screening rate was highest at six months and declined over time. | NR | NR | Among patients who made at least two visits, hypertension control was 13% at six months and plateaued around 25% at 42 months. |
| Zou, 2020 [38] | Sierra Leone | Quasi-experimental | Health workforce (TS), Medical products and technology, and leadership and governance | NR | 12 mmHg decline in Diastolic BP from 98 mmHg at BL in three months (t = 4.069, P = 0.001). No statistically significant decline in Systolic BP (15 mmHg from 172 mmHg; t = 1.701, p = 0.106). | NR | NR |

**Abbreviations: BP**—Blood Pressure; **BL**—Baseline; **TS** -Taskshifting strategy; **HIP**—Health Insurance Premium; **RCT**—Randomized Controlled Trial; **QI**—Quality Improvement; **NR**—Not reported; **CI**—Confidence Interval; **aOR**—Adjusted Odds Ratio

of leadership and governance–the collaborations with government and non-government, local and international organizations to train the trainers of hypertension care. The consortium provided training to the providers at the referral health facilities who, in turn, trained and gave supportive mentorship to the health facilities in the lower levels of the health system [38]. This intervention showed a significant reduction in diastolic (from 98 to 86 mmHg, t: 4.069, p: 0.001) but not systolic BP.

**Interventions targeting multiple health systems factors.** The health systems interventions on hypertension targeted two or more health system components in ten of the twelve

studies included. The health workforce and health information systems were the most common combination of factors targeted, followed by the health workforce and medical products. One randomized controlled trial and one quasi-experimental study which targeted a single health system factor, health workforce, and health financing, respectively, did not yield a significant improvement in hypertension outcomes. In contrast, studies which targeted multiple health system factors were likely to demonstrate significant improvements in hypertension outcomes.

## Discussion

This systematic review assessed the health system interventions for hypertension awareness, initiation and adherence to hypertension treatment, and achievement of BP control in SSA. Our main findings are that studies to address health systems challenges of hypertension care are very limited in quality and number. The studies we identified addressed mostly the health workforce challenges, access to equipment and medicines, and health information systems. The studies addressing hypertension care delivery issues, such as decentralization of hypertension care and integration delivery, leadership and governance, and financing aspects of the health systems, were minimal. Overall, we found that interventions targeting multiple aspects of the health systems were more likely to show significant improvements in hypertension outcomes than interventions targeting solo health system components.

The healthcare provider's knowledge and adherence to hypertension treatment guidelines stood out as one of the most studied health system interventions on hypertension [37, 39, 40]. The capacity building among healthcare providers and the task sharing and shifting of hypertension management roles to vital professionals like nurses, pharmacists, and community health workers align with scientific evidence from developing and developed countries [41, 42]. For instance, the nurses' role in managing hypertension has evolved from merely measuring and monitoring BP to collaboratively detecting, diagnosing, and referring patients with hypertension and its complications [43, 44]. In many countries and settings, nurses prescribe or dispense anti-hypertensive medications, provide patient education and counseling to ensure medication adherence, and assume leadership roles in spearheading quality improvement projects for better management of hypertension [43]. An estimation of the global gap in clinic visits for hypertension care reported that 50% of LMICs and 86% of lower-income countries have a physician deficit even if patients were to make only three annual visits to the health facilities for their hypertension [45]. Neupane and colleagues recommend shifting some hypertension management tasks to non-physician clinicians to bridge that gap [45]. Interdisciplinary collaboration is the way forward for SSA to provide quality hypertension care to patients; however, regional and national policies must align with and support this practice.

The second health system component widely explored is the supply of essential medical products, including BP measurement devices and anti-hypertensive medications. It is intuitive to formulate the association between the availability of medical products and hypertension outcomes, yet the challenges that lead to the disruption in the supply chain of medical products are a complex issue to disentangle. The common approach addressed by the interventions was to procure the needed equipment and medications but not to fix the inherent challenges in the availability of those medical products and the supply chain–or attempt to identify loopholes in the process to recommend viable and context-specific solutions. A separate narrative synthesis has identified multiple challenges to the availability of medical products in Africa [46]. The challenges identified include limited pharmaceutical industrial power and high costs of raw materials, overdependence on developed countries for these products, poor supply chain systems, poor government financing, and lack of investment in supply chain research [46]. Unless

studies work to identify and address the root cause of the inadequate supply of medical products, short-term solutions are unsustainable and unhelpful in the rising healthcare needs.

WHO defines health information systems as collecting, storing, analyzing, and utilizing patient and healthcare delivery data to improve quality, research, and inform policy [22]. "If you can't measure it, you can't improve it" is a quote often attributed to Peter Drucker about the critical need for a system to continuously collect data and measure what you are trying to improve. The studies we included in the current review examined mainly the patient data collection and sharing between providers and between providers and the patients to improve hypertension outcomes. In many developed countries, health information systems have gone to incorporate technology that has rendered data generation, processing, analysis, visualization, and sharing seamless and less burdensome [47]. One major challenge and critique of health information systems in developed countries is the lack of unification of health information systems across healthcare organizations [47]. Countries in SSA and other developing countries can learn from developed countries and build their health information infrastructure in a unified framework to facilitate patient follow-up outside and across the health facilities, expedite health data reporting, and promote research to advance evidence-based practice, funding allocation, and policy. The health information system interventions identified in the current review did not find studies exploring patient registries to prevent loss to follow-up in hypertension management.

The cross-integration of hypertension services is a service delivery system-level intervention explored for managing hypertension. The advocacy for integrating hypertension and HIV care originated from the realization that many patients with HIV were presenting with hypertension and other non-communicable chronic conditions such as diabetes. HIV care outcomes [48, 49]. Sierra Leone [50], South Africa [51], Tanzania and Uganda [52], and Malawi [53] have reported outcomes of some form of HIV and hypertension care integration from the national reports and cohort studies. In those studies and reports, patients demonstrated improved HIV and hypertension outcomes. There are some reasons why taking lessons from the management of chronic infectious diseases could be effective in hypertension care. The management of chronic infectious diseases, HIV and Tuberculosis, in resource-constrained settings has, despite still-existing challenges, produced strong evidence and effective strategies for managing patients with such complicated chronic health conditions amid, just to name a few; insufficient healthcare professionals [54], inaccessibility of health services [55], and poor information management systems [56]. In addition to demonstrating that task sharing with and task shifting to non-physician healthcare providers is possible and effective [54, 57, 58], the process of managing HIV and Tuberculosis has also demonstrated ways to manage loss to follow up by the use of strong information management systems [56], decentralization of services (another form of delivery system design) close to the community [59], and collaboration with community health workers [54, 58]; all the strategies that could be well adapted to hypertension management. One of the information management system strategies that have been effective in HIV and Tuberculosis control and hypertension management in other low-income countries is the use of patient registries for patient follow-up [60–63]. We did not find studies exploring the service delivery strategy of decentralizing hypertension care services closer to the community.

Financing of health systems is another critical aspect of hypertension and other health services' accessibility and utilization. Studies in the current review have shown that addressing the cost-related barriers to care can contribute to helping patients with hypertension achieve better outcomes–the subsidization of health insurance premiums. Universal health coverage through government-subsidized insurance premiums has been advocated as a solution to prevent catastrophic spending on healthcare services and associated poor health outcomes [64–

66]. However, financing universal health coverage requires political buy-in and collaboration across sectors, local and international [65].

While leadership and governance are central to the success of any endeavor to improve the quality of care, their role and impact in advancing hypertension care in SSA are not widely researched, and the few existing studies are predominantly of poor quality [67]. Thornton highlighted the critical values of leadership commitment, willingness to fund care, strategic and creative local and foreign partnerships, and evidence-based guidelines as the key reasons why Botswana surpassed the United Nations' HIV management goals of 95% awareness, 95% on treatment, and 95% [55]. A 3-year campaign mobilization on the role of clinical leaders in hypertension management at various health systems in the United States achieved the goal of 80% BP control within seven months [68]. A scoping review of interventions to strengthen the health professionals' leadership in SSA found that the opportunities for leadership development in SSA are scarce, and those available are of poor quality and lack a consistent evaluation framework [67]. There is a need for more studies on strengthening leadership in healthcare and exploring its impact on health outcomes, including hypertension in SSA.

We found that interventions addressing multiple health system factors were more likely to report better hypertension outcomes than isolated investment in a single health system component. Health systems in most SSA countries, especially rural areas, are often frail in the multiple health system factors [69]. The inventions aiming to improve how those health systems manage a health condition such as hypertension should first assess health systems' readiness to identify the areas of focus for those interventions.

As the burden of hypertension increases globally, the investment required by health systems to manage hypertension is rising. Hypertension is significantly associated with adverse outcomes of Coronavirus Disease 2019 (COVID-19), the ongoing pandemic ravaging the entire world's health systems [70, 71]. Studies reported that hypertensive patients had almost three times the odds of mortality compared to people without hypertension [70]. The long-term post-COVID symptoms haunt the patients with hypertension who survive COVID-19 more than the general population [72]. For instance, patients with hypertension are twice as likely to have migraine-like headaches and 68% higher odds of poor sleep than people without hypertension [72]. Apart from the COVID-19 rationale, under the Sustainable Development Goal (SDG) 3 (Ensure healthy lives and promote well-being for all at all ages), target 3.4 aims to reduce by one-third the premature mortality from non-communicable diseases [73]. The management of hypertension, a significant risk factor for cardiovascular diseases, is a step closer to achieving target 3.4 of SDG 3.

Standard tools for exploring the health system determinants for specific diseases are scarce in the realm of hypertension care in Africa. There is a dire need for psychometric tools to explore health system readiness to provide quality hypertension care. The availability of those tools would open doors for researchers interested in hypertension care to gather quality data on health system determinants and render the evidence to policymakers to develop policies directed at shortfalls in care delivery to improve the health outcomes of patients with hypertension.

Future studies should investigate the different aspects of hypertension service delivery, especially interventions that make service accessible, the role of leadership and governance, the latest methods of health information system management including the use of electronic medical records for patient follow up, as well as multi-level approaches (i.e., individual, community, provider, health system) in the improvement of hypertension care. To improve the quality of the studies, the quality improvement projects aimed at testing the interventions in clinical settings and evaluative projects aimed at generating knowledge will have to be rigorously guided by the science of study design. Since healthcare settings are ever dynamic with many

interconnected components, the appropriate design has to be carefully identified allowing the balance between controlling the allocation of the interventions with prevention of contamination and bias and the pragmatism of the project in the research setting [74, 75]. The procedures and reasons for omitting standard practices of study design and implementation will, however, have to be acknowledged and adequately described. Additionally, while the current review explored interventions conducted in SSA, most studies were conducted in three countries (Cameroon, Kenya, Nigeria, and South Africa). Future studies should explore health systems interventions for hypertension care in other countries of SSA. Such studies could inform the much-needed local guidelines for hypertension care and patient follow-up.

### Strengths and limitations

The limitation of the current review is that we conducted the literature search in English. Even though we reviewed studies published in both English and French, the English search terms, could limit our ability to identify studies published in other languages. We were unable to explore the isolated effect of individual health system factors interventions, because many system interventions were delivered in a bundled approach. However, this review offers valuable insights as the first to explore the health systems interventions on hypertension care in SSA, applied rigorous methods to critique the quality of the studies, and reviewed the interventions using the WHO health systems framework [22].

### Conclusion

The literature exploring health system interventions for hypertension management in SSA is limited in volume and quality. The combination of multiple health system interventions was likely to result in better hypertension outcomes. More rigorously designed studies addressing the different aspects of the health systems and their impact on hypertension outcomes, especially service delivery, health information management, and leadership and management, are needed in SSA to inform practical hypertension control efforts.

### Supporting information

**S1 Checklist. PRISMA 2009 checklist.**
(DOC)

**S1 Fig. Joanna Briggs Institute quality assessment results.**
(TIFF)

**S1 Table. Search terms.**
(DOCX)

**S2 Table. Standards for Quality Improvement Reporting Excellence (SQUIRES).**
(DOCX)

### Acknowledgments

We acknowledge the support of Stella Seal, the Library Informationist who helped in searching the databases, Dr. Sarah Szanton and Dr. William E. Rosa for the contribution of ideas in project ideation and discussion of findings, and Dr. Emmanuel Uwiringiyimana for the assistance in the full-text screening.

## Author Contributions

**Conceptualization:** Samuel Byiringiro, Cheryl R. Himmelfarb.

**Data curation:** Samuel Byiringiro.

**Formal analysis:** Samuel Byiringiro.

**Methodology:** Samuel Byiringiro, Oluwabunmi Ogungbe, Yvonne Commodore-Mensah, Khadijat Adeleye, Fred Stephen Sarfo, Cheryl R. Himmelfarb.

**Supervision:** Yvonne Commodore-Mensah.

**Validation:** Oluwabunmi Ogungbe, Yvonne Commodore-Mensah, Khadijat Adeleye, Fred Stephen Sarfo, Cheryl R. Himmelfarb.

**Visualization:** Samuel Byiringiro, Oluwabunmi Ogungbe.

**Writing – original draft:** Samuel Byiringiro, Oluwabunmi Ogungbe, Yvonne Commodore-Mensah, Cheryl R. Himmelfarb.

**Writing – review & editing:** Samuel Byiringiro, Yvonne Commodore-Mensah, Khadijat Adeleye, Fred Stephen Sarfo, Cheryl R. Himmelfarb.

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
