## [Decision Letter · Decision Letter 0]

25 Apr 2023

Health Systems Interventions for Hypertension Management and Associated Outcomes in Sub-Saharan Africa: A Systematic Review

PGPH-D-23-00184

Dear Mr Byiringiro,

We are pleased to inform you that your manuscript 'Health Systems Interventions for Hypertension Management and Associated Outcomes in Sub-Saharan Africa: A Systematic Review' has been provisionally accepted for publication in PLOS Global Public Health.

Best regards,

Julia Robinson

Executive Editor

Reviewer Comments (if any, and for reference):

Reviewer's Responses to Questions

**Comments to the Author**

1. Does this manuscript meet PLOS Global Public Health’s publication criteria? Is the manuscript technically sound, and do the data support the conclusions? The manuscript must describe methodologically and ethically rigorous research with conclusions that are appropriately drawn based on the data presented.

Reviewer #1: Yes

2. Has the statistical analysis been performed appropriately and rigorously?

Reviewer #1: Yes

3. Have the authors made all data underlying the findings in their manuscript fully available (please refer to the Data Availability Statement at the start of the manuscript PDF file)?

Reviewer #1: Yes

4. Is the manuscript presented in an intelligible fashion and written in standard English?

Reviewer #1: Yes

5. Review Comments to the Author

Reviewer #1: The authors have summarized in exquisite detail the status of hypertension management in sub-Saharan Africa. They have found that the literature on health care interventions dealing with hypertension there is severely limited. Importantly, they outline a logical approach to addressing this deficiency in the medical literature. The manuscript provides very useful information for health care policy makers and providers in sub-Saharan Africa and low income countries worldwide. The manuscript is well written and deserves publication in its current form with a high priority.

6. PLOS authors have the option to publish the peer review history of their article (what does this mean?). If published, this will include your full peer review and any attached files.

**Do you want your identity to be public for this peer review?** For information about this choice, including consent withdrawal, please see our Privacy Policy.

Reviewer #1: No
